# Full latitudinal marine atmospheric measurements of iodine monoxide

Hisahiro Takashima[1,2], Yugo Kanaya[2], Saki Kato[1], Martina M. Friedrich[3],

Michel Van Roozendael[3], Fumikazu Taketani[2], Takuma Miyakawa[2], Yuichi Komazaki[2],

Carlos A. Cuevas[4], Alfonso Saiz-Lopez[4], and Takashi Sekiya[2]

[1] Faculty of Science, Fukuoka University, Fukuoka, Japan

[2] Japan Agency for Marine–Earth Science and Technology (JAMSTEC), Yokohama, Japan

[3] Belgian Institute for Space Aeronomy (BIRA-IASB), Brussels, Belgium

[4] Department of Atmospheric Chemistry and Climate, Institute of Physical Chemistry Rocasolano (CSIC), Madrid, Spain

*Correspondence to*: Hisahiro Takashima (hisahiro@fukuoka-u.ac.jp) and Yugo Kanaya (yugo@jamstec.go.jp)

**Abstract.** Iodine compounds destroy ozone ($O_3$) in the global troposphere and form new aerosols, thereby affecting the global radiative balance. However, few reports have described the latitudinal distribution of atmospheric iodine compounds. This work reports iodine monoxide (IO) measurements taken over unprecedented sampling areas from the Arctic to the Southern Hemisphere and spanning sea surface temperatures (SSTs) of approximately 0°C to 31.5°C. The highest IO concentrations were observed over the Western Pacific warm pool (WPWP), where $O_3$ minima were also measured. There, negative correlation was found between $O_3$ and IO mixing ratios at extremely low $O_3$ concentrations. This correlation is not explained readily by the "$O_3$-dependent" oceanic fluxes of photolabile inorganic iodine compounds, the dominant source in recent global-scale chemistry-transport models representing iodine chemistry. Actually, the correlation rather implies that "$O_3$-independent" pathways can be similarly important in the WPWP. The $O_3$-independent fluxes result in a 15% greater $O_3$ loss than that estimated for $O_3$-dependent processes alone. The daily $O_3$ loss rate related to iodine over the WPWP is as high as approximately 2 ppbv despite low $O_3$ concentrations of approximately 10 ppbv, with the loss being up to 100% greater than that without iodine. This finding suggests that warming SST driven by climate change might affect the marine atmospheric chemical balance through iodine–ozone chemistry.

## 1 Introduction

Halogens play an important role in tropospheric and stratospheric chemistry through the catalytic destruction of ozone ($O_3$), which affects the atmosphere's oxidizing capacity and the radiative balance of the Earth (Alicke et al., 1999; Koenig et al.,

2020; Read et al., 2008; Saiz-Lopez et al., 2012; Saiz-Lopez et al., 2014; Simpson et al., 2015). Iodine, particularly is potentially important in tropospheric chemistry because of its rapid reactions, although its concentration in the troposphere is low compared to that of chlorine and bromine. Iodine also forms aerosol particles; it can thereby affect the global radiative balance (O'Dowd et al., 2002; Sipila et al., 2016; Gómez-Martín et al., 2020; Baccarini et al., 2020; Gómez-Martín et al., 2021; He et al., 2021).

Because of their low concentrations in the atmosphere, iodine compounds are difficult to quantify. Few reports have attempted to clarify their regional to global-scale sources and roles in atmospheric chemistry (Großmann et al., 2013; Mahajan et al., 2012; Prados-Roman et al., 2015a; Dix et al., 2013; Volkamer et al., 2015). In the past, the primary source of iodine in the troposphere has long been regarded as organic compounds in coastal areas (Davis et al., 1996; Carpenter et al., 2012; Prados-Roman et al., 2015a). However, results of recent studies suggest that iodine compounds over the open ocean are emitted

from inorganic sources following $O_3$ deposition over the ocean surface (Carpenter et al., 2013; Macdonald et al., 2014; Prados-Roman et al., 2015a). The inorganic sources are now regarded as the dominant emission term over the oligotrophic oceans in the global-scale chemistry-transport models representing iodine chemistry (e.g., Saiz-Lopez et al., 2014; Sekiya et al., 2020), although the emission process of inorganic iodine is still insufficiently clear in more recent studies (e.g., Inamdar et al., 2020).

This study specifically examines iodine monoxide (IO) in the marine boundary layer over the open ocean from the wide

latitudinal bands. Specifically, we examine processes occurring over the tropical western Pacific, where the global sea surface temperature (SST) reaches a maximum (warm pool) and where $O_3$ minima have been reported (Rex et al., 2014; Kanaya et al., 2019; Kley et al., 1996). Actually, IO observations in environments with SSTs of >30°C are limited. Observations of IO have been made in the tropics, but only for short time periods with SST > 30°C, if any (Großmann et al., 2013; Dix et al., 2013; Prados-Roman et al., 2015a). Although the importance of halogen chemistry as a driver of $O_3$ losses in this region has been

suggested (Großmann et al., 2013; Koenig et al., 2017), this point has yet to be examined in the context of full latitudinal distributions.

The initial production of atmospheric inorganic iodine species has not been fully examined in an environment where extremely low $O_3$ concentrations (<10 ppb) are observed. Over the Atlantic Ocean (in Cape Verde), long-term observations of iodine and ozone have been conducted, but they were in higher $O_3$ environments of approximately 20 ppbv (Read et al., 2008).

We therefore examined IO variations over the tropical western Pacific and their potential contributions to regional $O_3$ losses, with emphasis on SST as a potential key parameter controlling the initial iodine emissions. The global SST maximum is observed in the tropical western Pacific, but observations reported from earlier studies were only taken in the regions surrounding the maximum (Großmann et al., 2013; Prados-Roman et al., 2015a). Investigation of iodine variations in the tropics is also important for elucidating the stratospheric chemical balance (Koenig et al., 2017) because transport from the

troposphere to the stratosphere occurs through the Tropical Tropopause Layer (Takashima et al., 2008; Saiz-Lopez et al., 2015; Koenig et al., 2017; Holton et al., 1995). In fact, it is particularly important over the tropical western Pacific.

For this study, using the multi-axis differential optical absorption spectroscopy (MAX–DOAS) remote sensing technique, IO observations were made to quantify IO concentrations over the open ocean, covering the widest latitudinal range ever

examined with a single instrument. The technique uses scattered solar radiation at several elevation angles to obtain

atmospheric aerosol and gas profile concentrations (Hönninger et al., 2004; Wagner et al., 2004; Wittrock et al., 2004; Sinreich et al., 2005; Frieß et al., 2006; Kanaya et al., 2014). MAX–DOAS generally measures trace-gas contents over a long light path (up to 10–20 km) at low elevation angles. The long light path enables the detection of low concentrations of species of interest at near-surface altitudes. MAX–DOAS is therefore useful for quantifying low-abundance tropospheric trace gases such as IO over the open ocean.

Multi-platform measurements by MAX–DOAS from aircraft (Koenig et al., 2017; Volkamer et al., 2009) and ships (Großmann et al., 2013; Takashima et al., 2012; Volkamer et al., 2009) have been developed in recent years. Earlier studies have retrieved IO concentrations (typically < 1 pptv) in the marine boundary layer over the open ocean from shipboard MAX–DOAS measurements (Großmann et al., 2013; Mahajan et al., 2012; Prados-Roman et al., 2015a; Inamdar et al., 2020). Since 2008, the Japan Agency for Marine–Earth Science and Technology (JAMSTEC) has undertaken an unprecedented set of MAX–DOAS measurements on board the research vessels (R/Vs) *Kaiyo*, *Mirai*, and *Kaimei* around the world (Takashima et al., 2012). This report presents IO and $O_3$ variations over the open ocean from the Arctic to the Southern hemisphere observed on RV *Mirai* between 2014–2018.

## 2 Methodology

### 2.1 Iodine monoxide observations from ship-based MAX–DOAS measurements

The shipboard MAX–DOAS apparatus used for this study comprised two main components: an outdoor telescope and an indoor UV–Vis spectrometer (SP-2358; Acton Research Corp., coupled to a PIXIS-400B back-illuminated CCD detector; Teledyne Princeton Instruments). These were connected using a 10–14 m long fiber-optic cable (100 µm radius, 60-core or 40-core). The telescope unit was developed jointly by the Japan Agency for Marine–Earth Science and Technology (JAMSTEC) and PREDE Co. Ltd. (Tokyo, Japan). The movable prism of the telescope unit rotates for elevation angles (ELs) of 3°, 5°, 10°, 30°, and 90°. The EL is changed every minute to observe scattered sunlight. The target EL is attained by adjusting the angle of the prism actively, and by considering the angle of the ship's roll (Takashima et al., 2016). The telescope line-of-sight was off the starboard side of the ship with a field of view of approximately 1.0°. The spectrometer was housed in an adiabatic plastic box with the temperature held constant at 35°C ± 0.1°C using a temperature controller (KT4; Panasonic Inc., Japan). The CCD was cooled to −70°C. The spectrometer was equipped with a 600 line mm$^{-1}$ grating at 300 nm. The slit width was 100 µm. The typical exposure time was 0.1–0.2 s.

Spectral data were selected for analysis when the EL was within ±0.5° of the target. Data were analyzed using the DOAS method (Platt and Stutz, 2008). A nonlinear least-squares spectral fitting procedure was used to derive differential slant column densities (DSCDs) of the oxygen collision complex ($O_2$–$O_2$ or $O_4$) and IO using the QDOAS software package (Danckaert et al., 2017), for which absorption cross-section data presented in Table 1 were used. For $O_4$ and IO retrievals, 425–490 nm and 415–438 nm fitting windows were applied, respectively. Examples of fitting results and the time series of DSCDs are presented

respectively in Figures 1 and 2. The typical fitting error of the IO DSCDs was approximately $1 \times 10^{12}$ molecules cm$^{-2}$, with a detection limit of approximately $4 \times 10^{12}$ molecules cm$^{-2}$ ($2\sigma$).

The Mexican Maxdoas Fit (MMF) retrieval algorithm (Friedrich et al., 2019) was used for retrieval of IO profiles and vertical column densities. The version of MMF used in this study is the same as used in Frieß et al. (2019) and Tirpitz et al. (2021) but with adjusted a priori and variance-covariance matrix settings to fit for IO retrieval. MMF applies the optimal estimation method and uses a two-step approach in which the aerosol profile is first retrieved from O$_4$ DSCDs. Then, the IO profile is retrieved from the IO DSCDs using the earlier retrieved aerosol profile in the forward model. We used VLIDORT (v.2.7) (Spurr, 2006) as the forward model in a pseudo-spherical multiple-scattering setting. Only intensity information and its analytically calculated Jacobians were used. No other Stokes parameter was used. MMF was used in logarithmic retrieval space on a retrieval grid of up to 4 km with 200 m layer height.

Both *a priori* profiles were constructed as constant below 500 m with an exponentially decreasing profile above 500 m for aerosol and IO profiles to examine near-surface areas specifically. The *a priori* aerosol optical depth was set as 0.18. The *a priori* IO vertical column density (VCD) was set to $2.5 \times 10^{12}$ molecules cm$^{-2}$. The *a priori* covariance matrix $\mathbf{S_a}$ for both aerosol and IO retrieval was constructed using the square of 100% of the *a priori* profile on the diagonal and a correlation length of 200 m. For the aerosols, the only retrieved quantity was the partial aerosol optical depth per layer. Therefore, in the forward model, a constant single scattering albedo of 0.95 was used for both retrievals: aerosol and IO. The phase function moments were constructed using the Henyey–Greenstein phase function (Henyey and Greenstein, 1941) with a constant asymmetry factor of 0.72. The surface albedo in the forward models was set as 0.06. Here, the single scattering albedo, asymmetry factor, and surface albedo were used similarly to work presented by Großmann et al. (2013). The degrees of freedom (DOFs) for the IO retrieval for MR14-06 (leg1) were 1–1.4. It is also noteworthy that the observed IO contents might be a little low compared with those from earlier studies conducted over the open ocean because of inaccuracy of the water–vapor cross-section used in earlier retrievals (Lampel et al., 2015).

## 2.2 Zero-dimensional photochemical box model with iodine chemistry

A zero-dimensional photochemical box model (Kanaya et al., 2007a; Kanaya et al., 2007b) based on the Regional Atmospheric Chemistry Mechanism (RACM) (Stockwell et al., 1997) and custom iodine chemistry was updated to include 91 chemical species and 275 reactions (reactions of iodine chemistry added to RACM are presented in Table 3). It was used to simulate the time evolution of mixing ratios of O$_3$ (initially 18 ppbv) and iodinated species in the boundary layer with assumed height of 500 m over the equatorial Pacific region, where the maximum concentrations of IO and minimum concentrations of O$_3$ were observed. For O$_3$, dry deposition at a velocity of 0.04 cm s$^{-1}$ was considered (Pound et al., 2020). Entrainment flux of $1.2 \times 10^8$ molecules cm$^{-2}$ s$^{-1}$ was assumed for NO$_2$, for which the initial mixing ratio was assumed to be 15 pptv. Fluxes of hypoiodous acid (HOI) and I$_2$ from the ocean surface were estimated respectively (Carpenter et al., 2013) as $8.4 \times 10^7$ and $2.6 \times 10^6$ molecules cm$^{-2}$ s$^{-1}$ at 10 ppbv of O$_3$, for an aqueous I$^-$ concentration of 74 nM and wind speed of 5 m s$^{-1}$ ($8.9 \times 10^7$ molecules cm$^{-2}$ s$^{-1}$ as total HOI/I$_2$ (= HOI + 2I$_2$) flux). The I$^-$ concentration was referred from the nearest observation data at

12°N and 158°E (Tsunogai and Henmi, 1971). The assumed wind speed was from observations made during MR14-06 cruise over the region. For Case 1a, the fluxes were assumed to be linearly dependent on $O_3$, which is consistent with Carpenter et al. (2013). For Case 1b a 25% reduction of the flux was assumed, potentially because of the presence of sea-surface microlayer or dissolved organic matter (Hayase et al., 2010; Hayase et al., 2012; Shaw and Carpenter, 2013; Tinel et al., 2020). The blue band of Figure 3 represents the range of Cases 1a and 1b, representing the case with "$O_3$-dependent" fluxes. In Cases 2a and 2b, the $O_3$-dependent flux in Case 1a was reduced to half and compensated by "$O_3$-independent" inorganic iodine fluxes of 3.3 (or 6.6) $\times 10^7$ molecules $cm^{-2}$ $s^{-1}$ (red band of Figure 3, representing the "quasi-$O_3$-dependent" case). As a reference, a hypothetical case (Cases 3a and 3b) with purely "$O_3$-independent" flux of the magnitude of 9.9 (or 13) $\times 10^7$ molecules $cm^{-2}$ $s^{-1}$ was also tested (orange band of Figure 3, representing the "purely $O_3$-independent" case). The time-dependent simulations continued for five days with evaluation of the mixing ratio of $O_3$ and its relation with IO involving daytime averages (0600–1800 ship local time) over the first to fourth days. Dry deposition velocities of iodine species (I, IO, HI, HOI, OIO, $I_2O_2$, INO, $INO_2$, $IONO_2$, and $I_2$) were assumed to be 1 cm $s^{-1}$.

## 2.3 Backward trajectory calculation

The origins of airmasses over the tropical western Pacific were investigated using five-day backward trajectory calculations (Takashima et al., 2011) based on meteorological analysis data of the European Centre for Medium-range Weather Forecasts (ECMWF).

## 2.4 In situ gas measurements

For measurements of $O_3$ and CO, ambient air was sampled using approximately 20 m of Teflon tubing (6.35 mm outer diameter) from the bow (Kanaya et al., 2019). To avoid contamination from ship exhaust, 1-min data deviated more than $1\sigma$ from the hourly discrete average were deleted. The typical magnitude of $1\sigma$ over the remote ocean was approximately 0.1–0.5 ppbv. The $O_3$ and CO concentrations were measured respectively using UV and infrared absorptions with $O_3$ and CO monitors (49C and 48C; Thermo Scientific, USA). The $O_3$ instrument was calibrated twice per year in the laboratory, before and after deployment, using a primary standard $O_3$ generator. The CO instrument was calibrated on board twice per year, on embarking and disembarking of the instrument, using a premixed standard gas. The reproducibility of the calibration was to within 1% for $O_3$ and 3% for CO (Kanaya et al., 2019). The $O_3$ concentrations observed from the R/V *Mirai* cruises presented in Table 3 are shown in Figure S1.

## 3 Results and Discussion

The IO contents (differential slant column densities (DSCDs) for an elevation angle of 3°) observed from the R/V *Mirai* during seven research cruises during 2014–2018 are presented in Figure 4. The cruises are presented in Table 3. Although observations were limited to some seasons and years (e.g., Arctic measurements were limited to the Northern Hemisphere summer), whole

latitudinal bands were covered from 74°N to 67°S, and strong latitudinal variations of IO concentrations were observed, with a maximum detected clearly in the tropics (10°S – 10°N), but not at higher latitudes in either hemisphere. Over Southeast Asia (near Indonesia), high IO concentrations were sometimes observed near coastal areas. The highest values of up to approximately $2 \times 10^{13}$ molecules cm$^{-2}$ (DSCD) were also observed in the tropical western Pacific, with wide variations at global SST maxima (>30°C). From similar earlier studies (Gómez-Martín et al., 2013; Großmann et al., 2013; Mahajan et al., 2012) no data obtained under very high SST conditions over a long period were reported. Therefore, our IO observations at SST maxima (up to 31.5°C) and during more than two weeks represent the most comprehensive measurements of reactive iodine over the tropical Western Pacific warm pool (WPWP).

Specifically regarding IO variations over the tropical western Pacific, we found IO VCDs of approximately 0.7–1.8 × $10^{12}$ molecules cm$^{-2}$ (Figure 5). Five-day backward trajectories indicate that air masses in this region originated from the open ocean (Figure 5). The carbon monoxide (CO) content was constantly low (60 ppbv, Figure 6), which is also consistent with an air mass originating from the open ocean. In addition, the chlorophyll content, based on satellite MODIS measurements (NASA Level-3 ver. 2018) in the source region, was also low (Figure 5), implying that any organic source of iodine can be expected to be negligible (although we also must consider abiotic organic source as well as mesotrophic conditions (Jones et al., 2010)). The IO data collected over the tropical western Pacific are consistent with I$^-$ variations reported in earlier studies (Chance et al., 2014; Chance et al., 2019; Sherwen et al., 2016), indicating an increase of I$^-$ concentration with SST.

For the time series of IO concentrations near the ocean surface (0–200 m height, Figure 7), the values were approximately 0.3–0.8 pptv, with wide variations over a timescale of a few days. The IO concentration near the surface depends on the shape of the *a priori* profile used for the retrieval, but day-to-day variations near the surface were unaffected by the choice of profile. Insufficient data were retrieved to document diurnal IO variations accurately. At times, the O$_3$ concentrations were generally low (<20 ppbv) and extremely low (<10 ppbv) (Figure 7). One unique finding was that, even under low-O$_3$ conditions, negative correlation was found between IO and O$_3$ concentrations in the daily dataset (Figures 3 and 7). Laboratory studies indicate that high O$_3$ concentrations can cause emission of iodine from ocean to atmosphere (Carpenter et al., 2013; Macdonald et al., 2014; Sakamoto et al., 2009). This "O$_3$-dependent" iodine release has been regarded as being more dominant than other "O$_3$-independent" types of emission, including photo-labile iodocarbons such as CH$_2$I$_2$, CH$_2$ICl, and their subsequent photolysis over the open oceans in every global-scale chemistry transport model representing iodine chemistry (Saiz-Lopez et al., 2014; Sekiya et al., 2020; Sherwen et al., 2016). However, with an "O$_3$-dependent" HOI/I$_2$ flux of approximately $9 \times 10^7$ molecules cm$^{-2}$ s$^{-1}$ (section 2.2), as expected under an O$_3$ mixing ratio of approximately 10 ppbv, a zero-dimensional box model was not able to reproduce the negative correlation found between IO and O$_3$. Because the initial HOI/I$_2$ release flux limited by O$_3$ in the <12 ppbv mixing-ratio range cannot drive the strong O$_3$ reduction, the scenario produced only a positive correlation (Case 1, Figure 3). In contrast, another case in which the "O$_3$-independent" flux was added to compensate for the "O$_3$-dependent" term weakened by a factor of 2 (Case 2, Figure 3) better reproduced the observed trend. The weakened flux might be explained by dissolved organic carbon (Shaw and Carpenter, 2013) or the presence of a sea-surface microlayer (Tinel et al., 2020) impeding iodine vaporization. The added "O$_3$-independent" flux is not explainable solely by flux from photolyses of

iodocarbons within the marine boundary layer (approximately $10^7$ molecules cm$^{-2}$ s$^{-1}$) generally assumed in the three-dimensional models (Saiz-Lopez et al., 2014; Sekiya et al., 2020; Sherwen et al., 2016). While indirectly considering the global total fluxes of CH$_2$IX (X = I, Br, Cl) as described by Ordóñez et al. (2012) in these model simulations, the "Chl-a-based" parameterization reduced the fluxes to too-low levels over this oceanic region. It therefore necessitates a survey of missing sources. The third case, with only an "O$_3$-independent" flux (Case 3, Figure 3) might explain the negative correlation more easily, whereas the total change of the flux type not being simply supported. We therefore hypothesize that O$_3$-independent processes are more important than has been represented by recent models. Indeed, a larger magnitude of organic iodine flux (approximately $7 \times 10^7$ molecules cm$^{-2}$ s$^{-1}$) was reported in the low-latitude Pacific (Großmann et al., 2013), and would therefore be the most likely cause of the negative correlation. However, that study (Großmann et al., 2013) relied on assumption of an even larger inorganic iodine emission flux to explain the observed IO concentrations. Therefore, our analysis is the first to suggest that the "O$_3$-independent" flux can be comparably important to the "O$_3$-dependent" flux in this region. Other O$_3$-independent iodine release mechanisms such as photooxidation of aqueous I$^-$ (Watanabe et al., 2019) might also be worth exploring. The modelled net O$_3$ loss rate attributable to iodine in Case 2 increased by up to 100% over that without iodine. The O$_3$ loss rate in the iodine cycle in Case 2 increased by approximately 15% over that in Case 1 (Table 4).

The expectation that a positive correlation between O$_3$ and IO would occur with O$_3$-dependent processes over a low O$_3$ concentration range was also confirmed using three-dimensional global chemistry-transport models including halogen chemistry (Sekiya et al., 2020; Saiz-Lopez et al., 2014) over the tropical western Pacific (Figures S2, S3). An alternative explanation for the observed negative correlation would be the mixing of air masses with different degrees of iodine chemistry. If so, such negative correlation could appear in the chemistry-transport model results. However, this feature was not found. Therefore, we propose an "O$_3$-independent" flux. Over the Atlantic, the O$_3$ mixing ratio rarely reaches these low levels (10 ppbv or less). Therefore, such process analyses have not been undertaken there. Under the influence of "O$_3$-independent" sources, even lower O$_3$ concentrations would be attainable. Radiative forcing of O$_3$, as estimated recently with halogen chemistry (Sherwen et al., 2017; Iglesias-Suarez et al., 2020; Saiz-Lopez et al., 2012; Hossaini et al., 2015), might be influenced by the dependence of iodine flux on O$_3$ concentration, which might play a major role in estimating past and future concentrations of O$_3$.

The time series of meteorological parameters including wind speed and SST was also investigated, but clear correlation such as that shown by O$_3$ and IO was not observed, in the correlation with O$_3$ or IO concentrations on a timescale of a few days (Figure 6). An earlier study (Kanaya et al., 2019) investigating the diurnal variation of O$_3$ in this area based on a comparison of observational data and a chemical transport model indicated that an as-yet-unidentified O$_3$ loss might occur over the tropical western Pacific. Our results imply that iodine chemistry plays an important role in O$_3$ loss in the area of SST maxima, which is regarded as an entry point from the troposphere to stratosphere. Moreover, these results provide insights into the manner by which increasing SST associated with climate change might modify the marine atmospheric chemical balance, which warrants further investigation. Results of recent studies indicate a roughly threefold increase in iodine since the 1950s, with at least 50% attributed to anthropogenic O$_3$ (Cuevas et al., 2018; Legrand et al., 2018; Zhao et al., 2019). If half of the

inorganic flux were $O_3$-independent, as suggested by Case 2, then either some other cause should be sought, or the change in $O_3$-dependent fluxes to produce the observed change is even more dramatic than previously thought. Further investigation of these points is necessary.

## 4. Summary

In this study, shipboard multi-axis differential optical absorption spectroscopy (MAX–DOAS), a remote sensing technique, was used during seven research cruises covering the widest latitudinal bands from the Arctic to the Southern Hemisphere as ever made with a single instrument, spanning SSTs of approximately 0°C to 31.5°C, allowing investigation of the variation of IO concentrations. It was particularly abundant over the tropical western Pacific (warm pool), appearing as an "iodine fountain", where SST maxima (>30°C) and $O_3$ minima are observed.

This report describes negative correlation between IO and $O_3$ concentrations over the IO maximum, even under extremely low $O_3$ conditions, which few earlier studies have demonstrated. This correlation is not explained easily by the $O_3$-dependent oceanic fluxes of photolabile inorganic iodine compounds adopted for recent simulation studies. Our findings rather imply that "$O_3$-independent" pathways which release iodine compounds from the ocean are also important. Iodine input to the atmosphere from the ocean surface is greater in areas of higher SST, leading to an "iodine fountain" in the Western Pacific warm pool because the $I^-$ concentration in the ocean surface is likely to be higher in these areas. This higher concentration might contribute to more pronounced $O_3$ destruction over the Western Pacific warm pool than estimated earlier. Warming SSTs associated with climate change can change the atmospheric chemical balance through halogen chemistry, warranting further quantitative investigation.

**Table 1:** Cross-sections of iodine monoxide (IO) and $O_4$ differential slant column densities used for this study

|  | Component | Reference |
|---|---|---|
| IO | $NO_2$ | Vandaele et al. (1998) |
|  | $O_3$ | Bogumil et al. (2000) |
|  | $H_2O$ | HITEMP* (Rothman et al., 2013) |
|  | IO | Gómez-Martín et al. (2005) |
| $O_4$ | $NO_2$ | Vandaele et al. (1998) |
|  | $O_3$ | Bogumil et al. (2000) |
|  | $H_2O$ | HITEMP (Rothman et al., 2013) |
|  | $O_4$ | Thalman and Volkamer (2013) |

\* Correction factors from Lampel et al. (2015) were applied.

**Table 2:** Reactions of iodine chemistry added to RACM

| Reactants | Products | A (cm$^3$ molecule$^{-1}$ s$^{-1}$) | Ea/R (K) | Reference |
|---|---|---|---|---|
| I + O$_3$ | IO + O$_2$ | $2.10 \times 10^{-11}$ | 830 | Sherwen et al. (2016) |
| I + HO$_2$ | HI + O$_2$ | $1.50 \times 10^{-11}$ | 1090 | Sherwen et al. (2016) |
| IO + NO | I + NO$_2$ | $7.15 \times 10^{-12}$ | −300 | Sherwen et al. (2016) |
| IO + HO$_2$ | HOI + O$_2$ | $1.40 \times 10^{-11}$ | −540 | Sherwen et al. (2016) |
| IO + IO | 0.43OIO + 0.71I + 0.43I$_2$O$_2$ | $9.60 \times 10^{-11}$ | 0 | Stutz et al. (1999) |
| OH + HI | I + H$_2$O | $1.60 \times 10^{-11}$ | −440 | Sherwen et al. (2016) |
| HOI + OH | IO + H$_2$O | $5.00 \times 10^{-12}$ | 0 | Sherwen et al. (2016) |
| I + NO$_3$ | IO + NO$_2$ | $4.50 \times 10^{-10}$ | 0 | Mcfiggans et al. (2000) |
| IO + CH$_3$O$_2$ | 0.25I + 0.254HCHO + 0.25HO$_2$ + 0.75HOI + 0.746ORA1 + 0.004H$_2$O$_2$ | $1.00 \times 10^{-11}$ | 0 | Stutz et al. (1999) |
| IO + $h\nu$ | I + O$_3$ | | | Kanaya et al. (2007b) |
| HOI + $h\nu$ | I + OH | | | Kanaya et al. (2007b) |
| INO$_2$ + $h\nu$ | 0.5I + 0.5NO$_2$ + 0.5IO + 0.5NO | | | Kanaya et al. (2007b) |
| IONO$_2$ + $h\nu$ | 0.5IO + 0.5NO$_2$ + 0.5I + 0.5NO$_3$ | | | Kanaya et al. (2007b) |
| OIO + OH | HOI + O$_2$ | $7.00 \times 10^{-12}$ | 0 | Kanaya et al. (2007b) |
| I$_2$O$_2$ + $h\nu$ | I + OIO | | | Kanaya et al. (2007b) |
| I + NO$_2$ (+M) | INO$_2$ (+ M) | $5.40 \times 10^{-12}$ | 0 | Kanaya et al. (2003) |
| INO$_2$ | I + NO$_2$ | $9.94 \times 10^{+17}$ | 11859 | Sherwen et al. (2016) |
| IO + NO$_2$ (+M) | IONO$_2$ (+ M) | $3.70 \times 10^{-12}$ | 0 | Kanaya et al. (2003) |
| IONO$_2$ | IO + NO$_2$ | $2.10 \times 10^{+15}$ | 13670 | Sherwen et al. (2016) |
| I + NO (+M) | INO (+ M) | $4.10 \times 10^{-13}$ | 0 | Kanaya et al. (2003) |
| INO | I + NO | $1.40 \times 10^{-1}$ | 0 | Kanaya et al. (2003) |
| OIO + NO | IO + NO$_2$ | $1.10 \times 10^{-12}$ | −542 | Sherwen et al. (2016) |
| IO + ISOP | 0.25I + 0.132MACR + 0.855OLT + 0.25HO$_2$ + 0.179HCHO + 0.75HOI + 0.075H$_2$O$_2$ + 0.9OH | $1.00 \times 10^{-11}$ | 0 | Kanaya et al. (2007b) |
| IBr + $h\nu$ | I | | | Kanaya et al. (2007b) |
| OIO + $h\nu$ | I + O$_2$ | | | Kanaya et al. (2007c) |
| I$_2$ + OH | HOI + I | $2.10 \times 10^{-10}$ | | Sherwen et al. (2016) |
| I$_2$ + NO$_3$ | I + IONO$_2$ | $1.50 \times 10^{-12}$ | | Sherwen et al. (2016) |
| I$_2$ + $h\nu$ | 2I | | | Alicke et al. (1999) |
| IO + OIO | I$_2$O$_3$ | $1.5 \times 10^{-10}$ | | Sherwen et al. (2016) |
| OIO + OIO | I$_2$O$_4$ | $1.5 \times 10^{-10}$ | | Sherwen et al. (2016) |
| I$_2$O$_2$ | OIO + I | 1.13 | | Saiz-Lopez et al. (2016) |
| I$_2$O$_2$ | IO + IO | 0.00532 | | Saiz-Lopez et al. (2016) |

| | | | | |
|---|---|---|---|---|
| I$_2$O$_4$ | OIO + OIO | 0.0879 | | Saiz-Lopez et al. (2016) |
| HOI + NO$_3$ | IO + HNO$_3$ | $2.7 \times 10^{-12} \times (300/T)^{2.66}$ | | Saiz-Lopez et al. (2016) |
| I$_2$O$_3$ + $hv$ | IO + OIO | | | Saiz-Lopez et al. (2016) |
| I$_2$O$_4$ + $hv$ | OIO + OIO | | | Saiz-Lopez et al. (2016) |

**Table 3:** Research cruises of the R/V *Mirai* that generated data used for this study

| Cruise | Period | Area |
|---|---|---|
| MR14-06 (leg 1) | 8 Nov – 3 Dec, 2014 | Western Pacific, Tropics |
| MR15-04 | 6 Nov – 21 Nov, 2015 | Western Pacific, East Indian Ocean |
| MR15-05 (leg 2) | 14 Jan – 24 Jan, 2016 | Western Pacific |
| MR16-06 | 24 Aug – 4 Oct, 2016 | Arctic Ocean, Bering Sea, North Pacific |
| MR16-09 (leg 3) | 8 Feb – 3 Mar, 2017 | Southern Ocean |
| MR17-05C | 25 Aug – 29 Sep, 2017 | Arctic Ocean, Bering Sea, North Pacific |
| MR17-08 | 22 Nov, 2017 – 17 Jan 2018 | Western Pacific, East Indian Ocean |

**Table 4.** Net and process-specific O$_3$ loss rates in three cases at an O$_3$ concentration of 10 ppbv, as calculated using the box model

| | IO [ppbv] | net loss [ppbv d$^{-1}$] | HOx/Ox cycle loss [ppbv d$^{-1}$] | Iodine cycle loss [ppbv d$^{-1}$] |
|---|---|---|---|---|
| w.o. iodine | 0 | −1.06 | −1.64 | 0 |
| Case 1 | 0.553–0.741 | −1.85 – −2.08 | −1.62 | −0.519 – −0.720 |
| Case 2 | 0.611–0.851 | −1.92 – −2.21 | −1.62 | −0.579 – −0.844 |
| Case 3 | 0.723–0.960 | −2.05 – −2.34 | −1.62 | −0.700 – −0.967 |

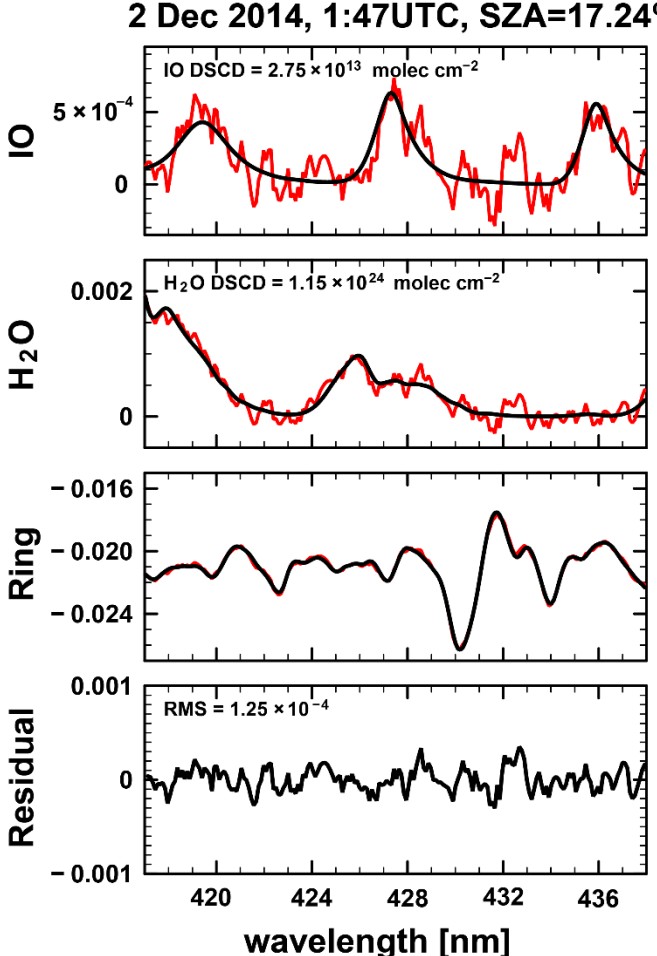

**Figure 1:** Nonlinear least-squares spectral fitting results for IO concentrations observed on 2 December 2014. The top two panels show fitting for IO and $H_2O$. Black lines represent the cross-section scaled to the spectrum (red) determined by differential optical absorption spectroscopy. The lower two panels show the Ring-effect contribution and the residual spectrum.

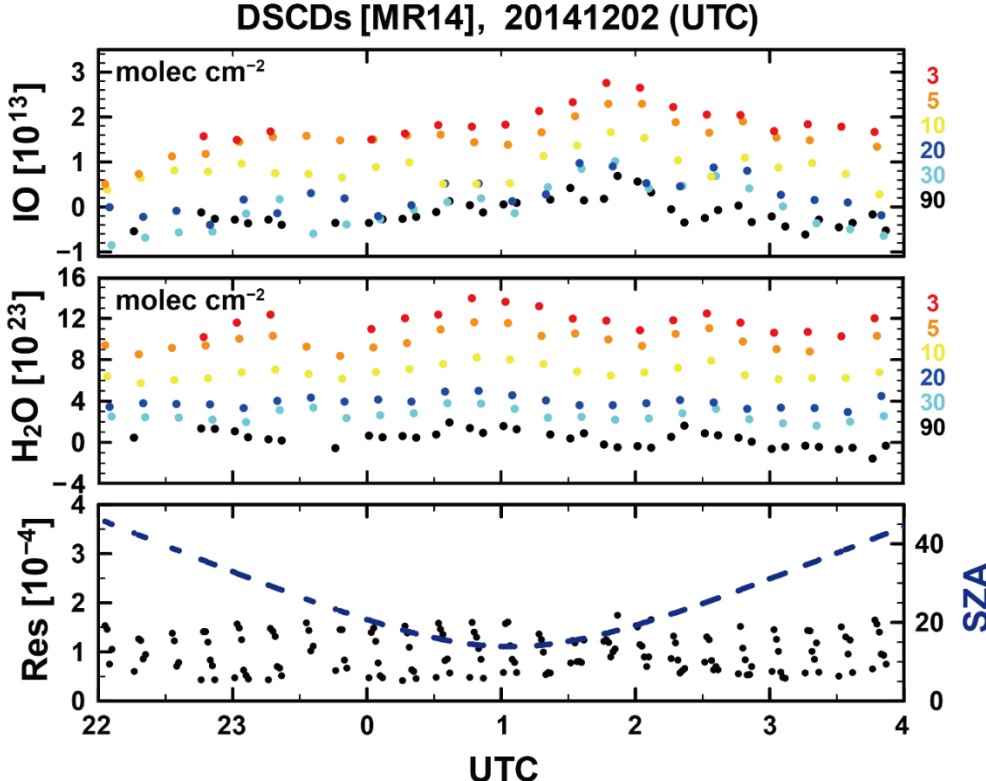

**Figure 2:** Time series of IO and $H_2O$ differential slant column densities (DSCDs) for elevation angles of 3°, 5°, 10°, 20°, 30°, and 90°; RMS residual; and the solar zenith angle observed on 1–2 December 2014 over the tropical western Pacific.

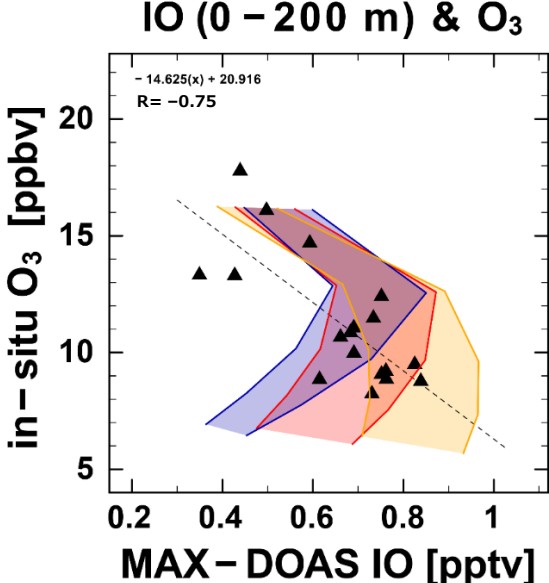

**Figure 3:** Daily median IO mixing ratio for 0–200 m (pptv) observed by MAX–DOAS versus daily median *in situ* ozone mixing ratio (ppbv). Results of box-model simulations with "O₃-dependent" (Case 1), "quasi-O₃-dependent" (Case 2), and "pure O₃ independent" (Case 3) emission fluxes of iodine compounds are superimposed respectively as blue-, red-, and orange-shaded areas.

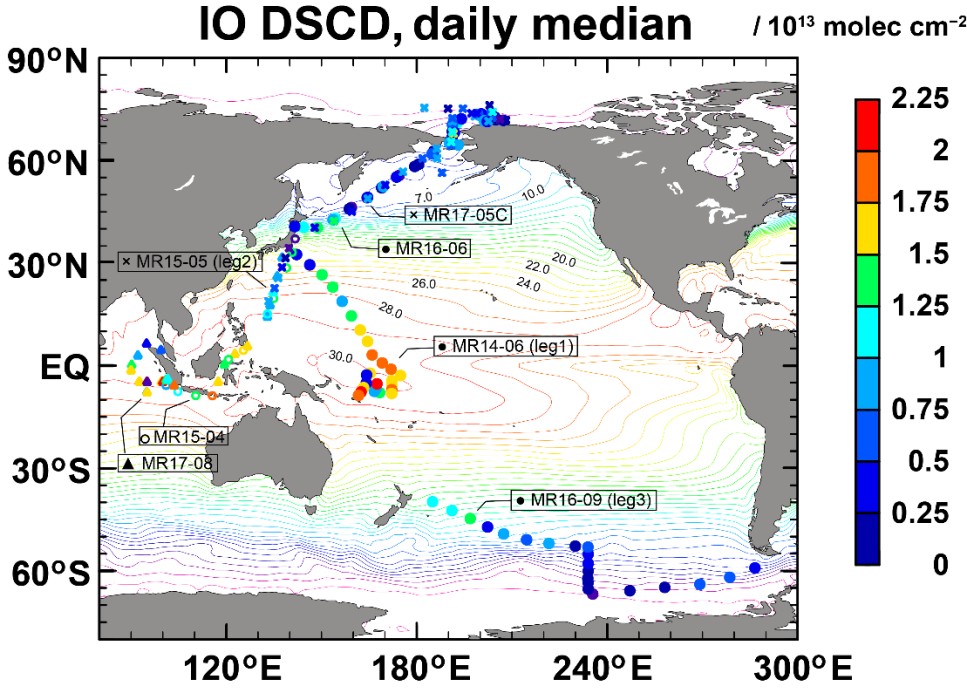

**Figure 4:** Daily median IO content (differential slant column densities (DSCD) for an elevation angle of 3°; molecules cm$^{-2}$) observed from the R/V *Mirai* during 2014–2018. Color contours represent the optimum interpolated SST averaged for 2014–2018.

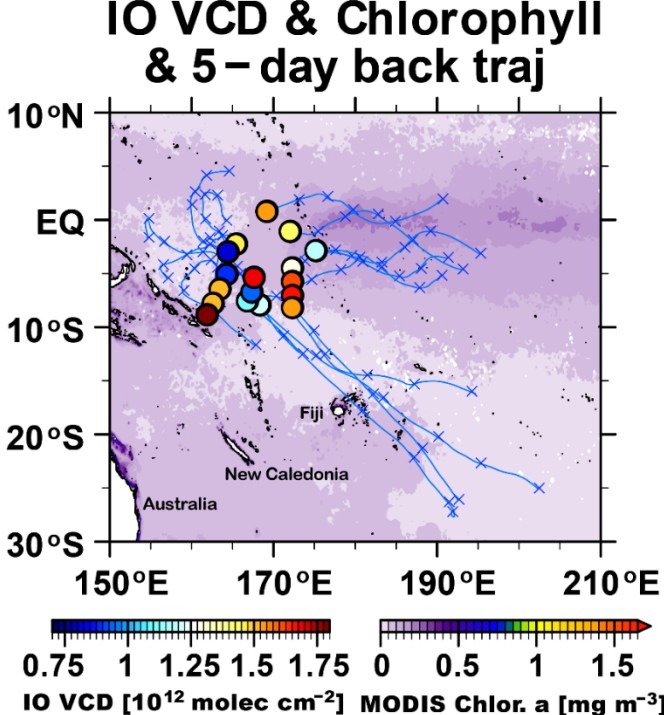

**Figure 5:** Daily median tropospheric IO vertical column densities (VCDs, molecules cm$^{-2}$) observed from the R/V *Mirai* during 16 November to 2 December 2014 and chlorophyll-a concentrations observed via satellite (MODIS). Blue crosses and lines represent five-day backward trajectories.

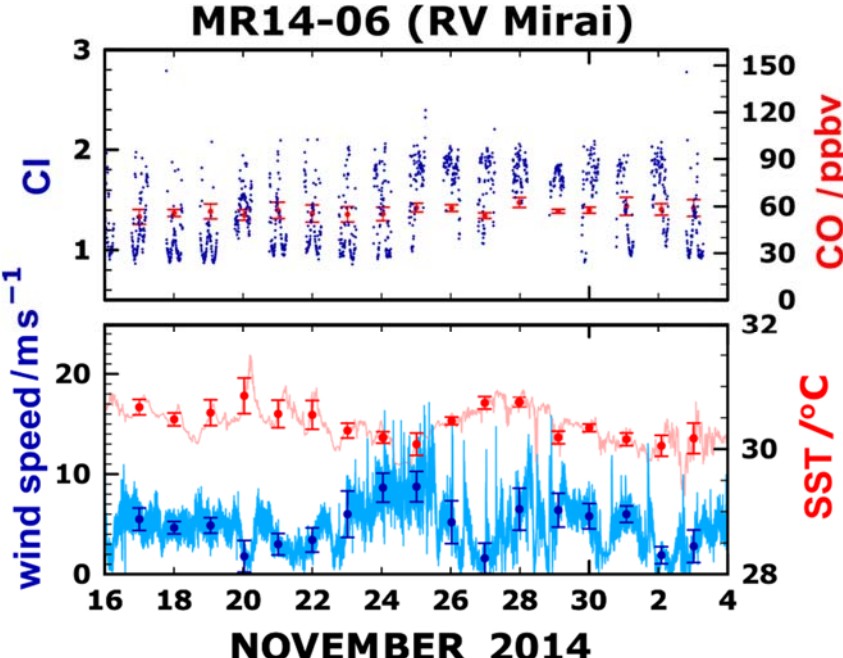

**Figure 6:** Time series of the CO mixing ratio [ppbv], color index (CI; defined as the ratio of the measured intensities at the two wavelengths of 500 and 380 nm (Takashima et al., 2009)), wind speed (m s$^{-1}$), and SST (°C).

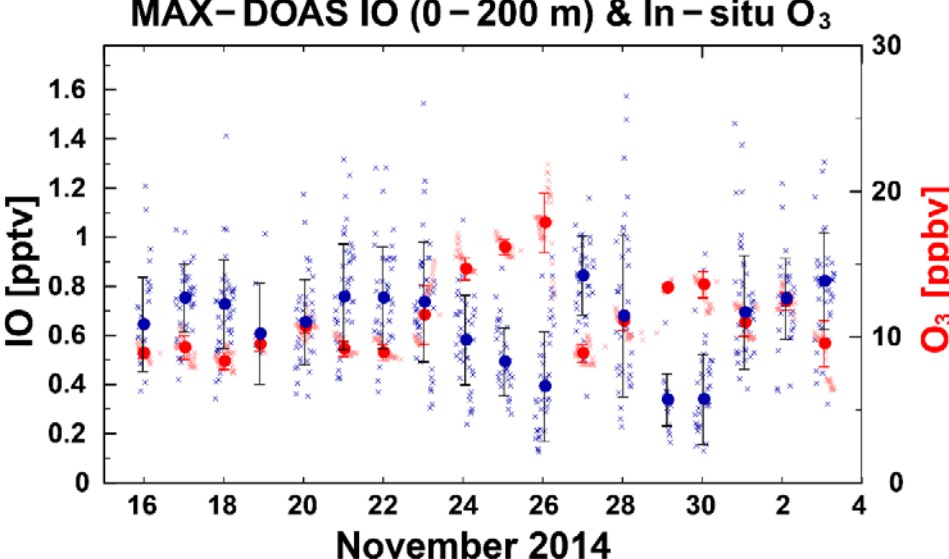

**Figure 7:** Time series of IO mixing ratio for 0–200 m (blue, pptv) observed by MAX–DOAS, and *in situ* O₃ mixing ratio (red, ppbv). Circles and horizontal bars respectively represent the  daily median and one standard deviation.

**Author contributions**

HT designed the study, conducted shipboard MAX–DOAS observations and analyses, and wrote the manuscript. YKa proposed the research concept, supported the MAX–DOAS observations, and conducted $O_3/CO$ observations and 0-D box model calculations. KS supported the observations and analysis. MF conducted the retrieval of IO profiles and IO VCDs. MV supported the DOAS analysis. FT, TM, and YKo supported the MAX–DOAS observations. CAC, AS-L, and TS conducted a simulation using a global chemical model. All co-authors provided comments to improve the manuscript.

**Data availability**

MAX-DOAS data are available by contacting the corresponding authors. Other data is available at the following sites (DARWIN).

MR14-06 (leg 1): https://doi.org/10.17596/0001862

MR15-04: https://doi.org/10.17596/0001975

MR15-05 (leg 2): https://doi.org/10.17596/0002030

MR16-06: https://doi.org/10.17596/0001870

MR16-09 (leg 3): https://doi.org/10.17596/0000026

MR17-05C: https://doi.org/10.17596/0001879

MR17-08 (leg1): https://doi.org/10.17596/0001881

MR17-08 (leg2): https://doi.org/10.17596/0001882

**Supplement**

Supporting information accompanies this paper.

**Acknowledgments**

We thank K. Kruger, Y. Yamashita, and K. Hara for their useful comments. We also thank Robert Spurr for free use of the VLIDORT radiative transfer code package. DOAS analysis involved the QDOAS software. We used MODIS chlorophyll *a*, OI SST, and ECMWF meteorological data. Figures were produced using the GFD-Dennou Library. This work was supported in part by funding from Fukuoka University (Grant No. 197103). This study has also received funding from the European Research Council Executive Agency under the European Union's Horizon 2020 Research and Innovation programme (Project 'ERC-2016-COG 726349 CLIMAHAL').

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

**Figure S1:** Ozone mixing ratio [ppbv] observed from the R/V *Mirai* cruises presented in Table 1 during 2014–2018.


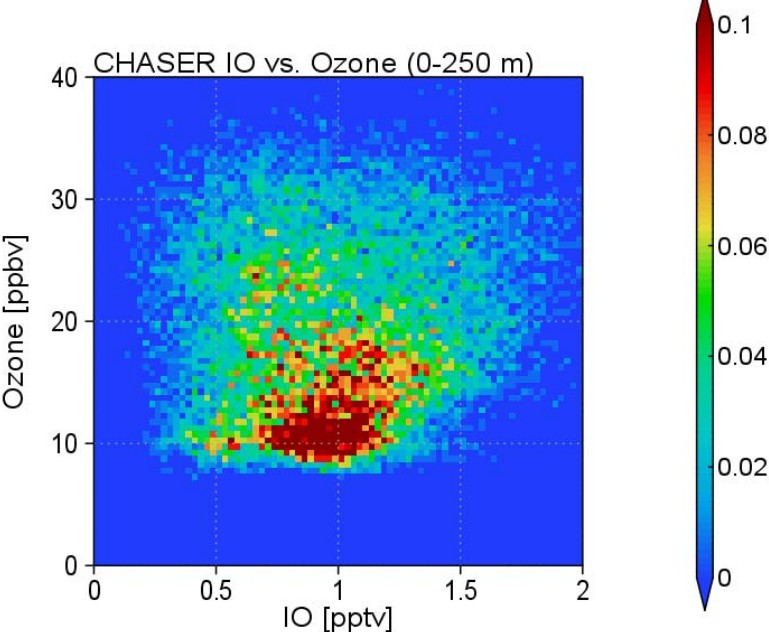

**Figure S2:** Two-dimensional histogram [%] as a function of IO volume mixing ratio [pptv] and ozone mixing ratio [ppbv] for 0–250 m
altitudes simulated using a global chemical model (Sekiya et al., 2020) during the observation period (Nov–Dec 2014) over the tropical
western Pacific (0–15°N, 150–165°E).

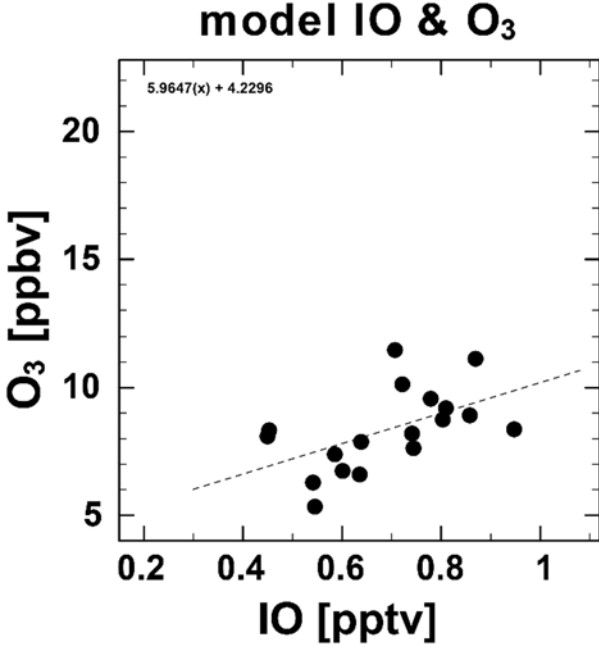

**Figure S3:** Scatterplot of IO mixing ratio [pptv] and ozone mixing ratio [ppbv] simulated by global chemical model (Saiz-Lopez et al., 2014) along the cruise track (MR14-06) over the tropical western Pacific.

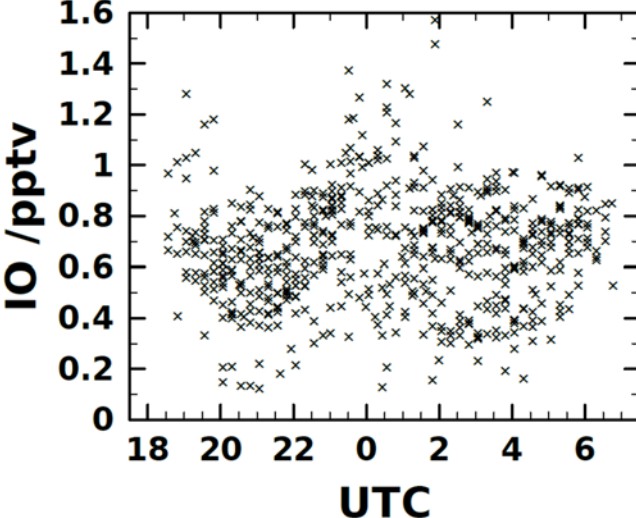


**Figure S4:** Diurnal variation of IO mixing ratio for 0–200 m [pptv] during the MR14-06 (leg1) cruise.

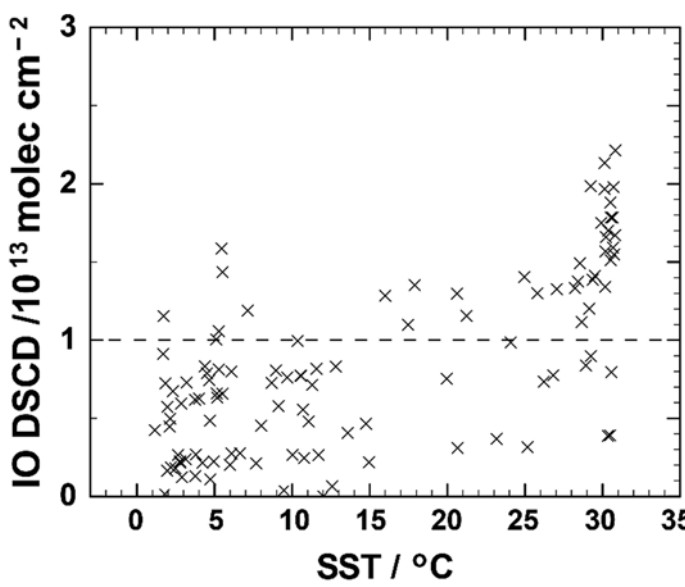


**Figure S5**: Scatter plot of SST [°C] and IO DSCD (el = 3°) [$10^{13}$ molecules cm$^{-2}$] over the remote ocean.

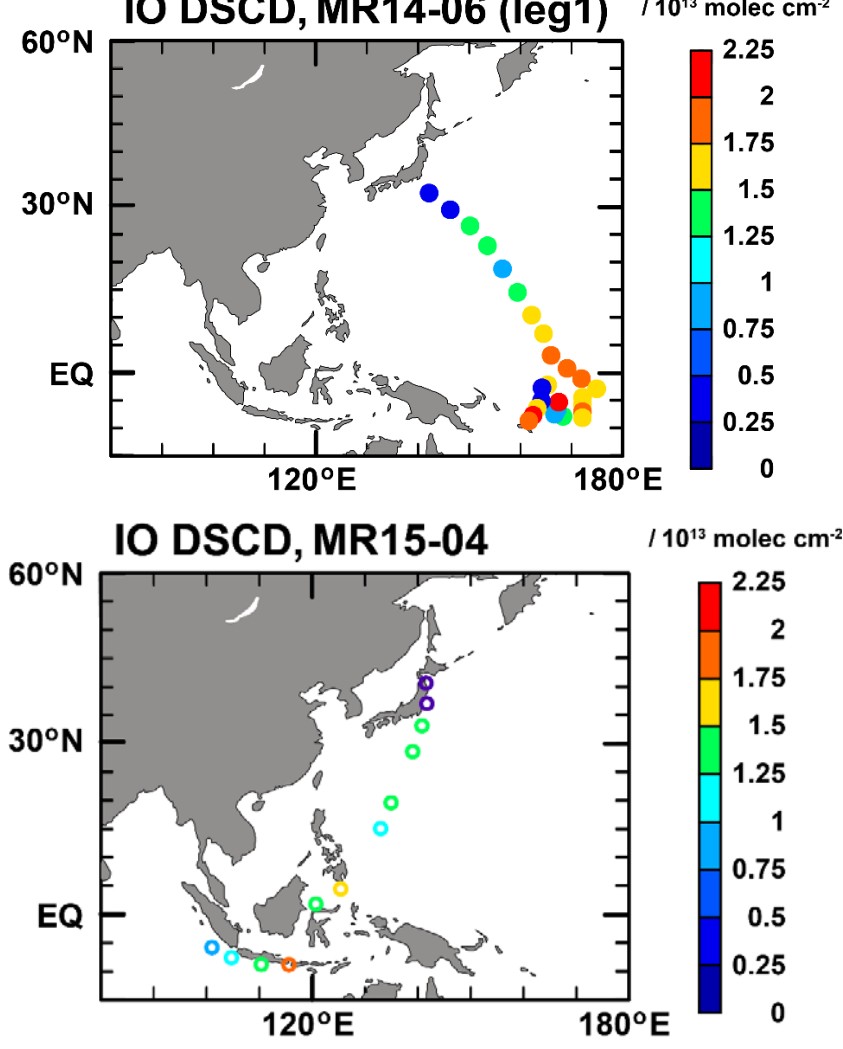

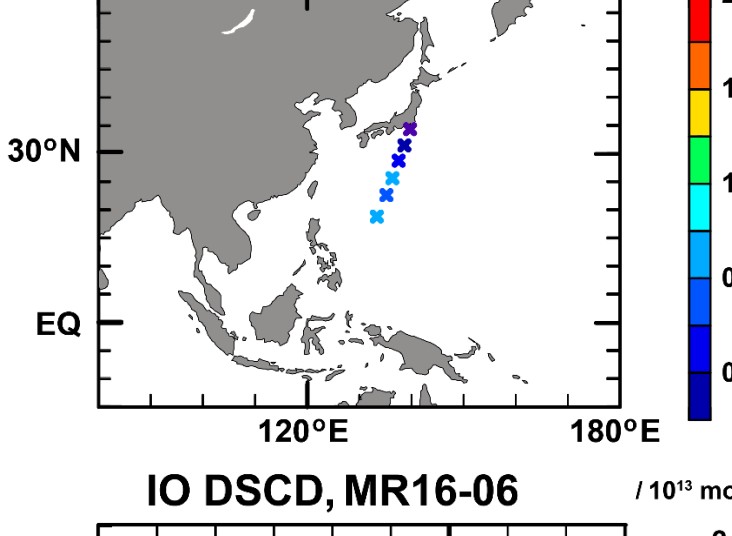

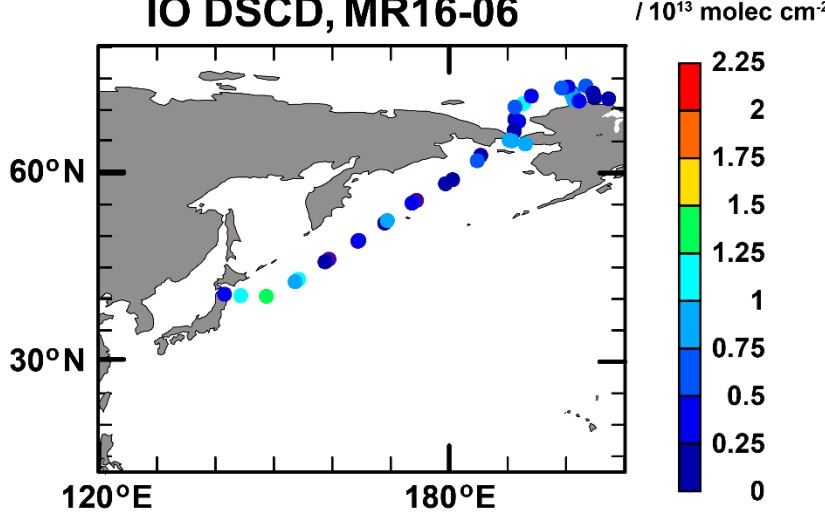


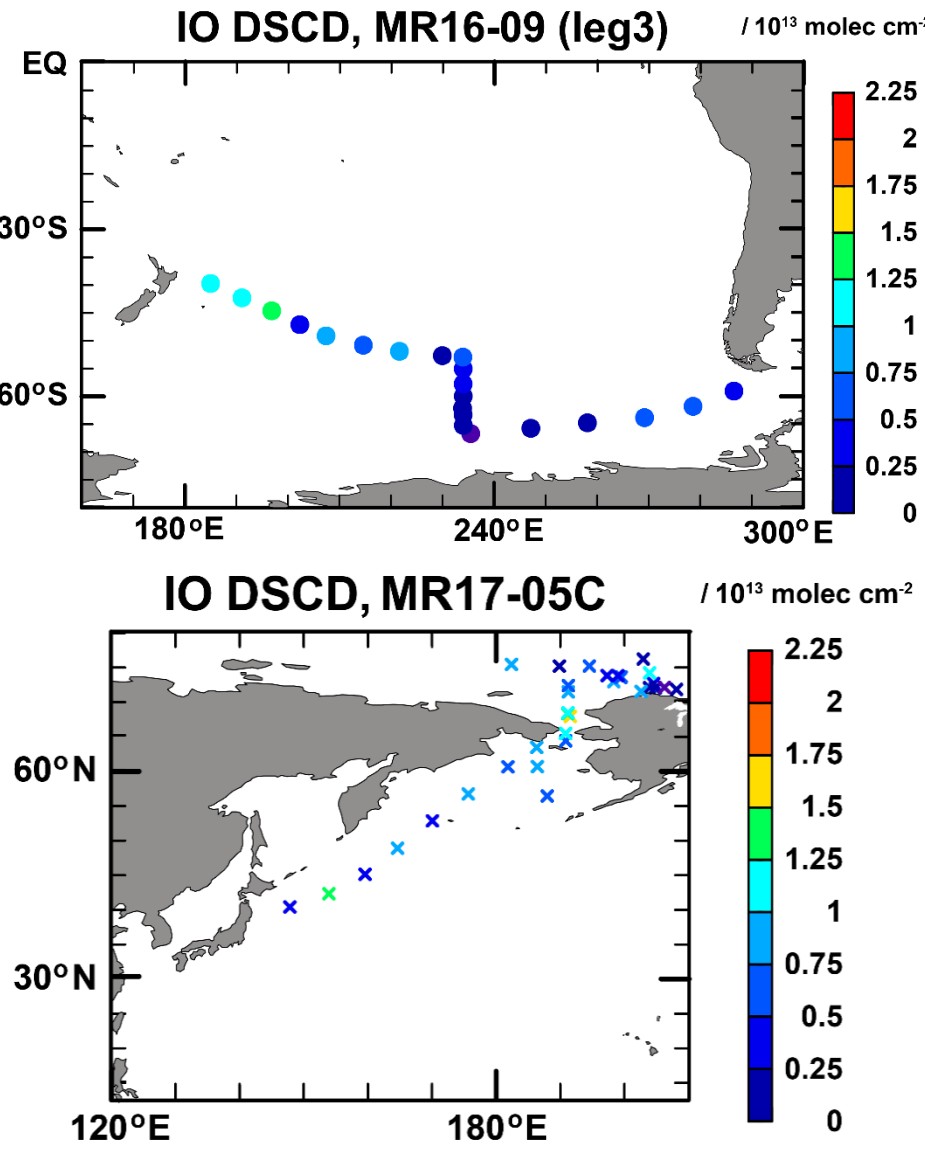

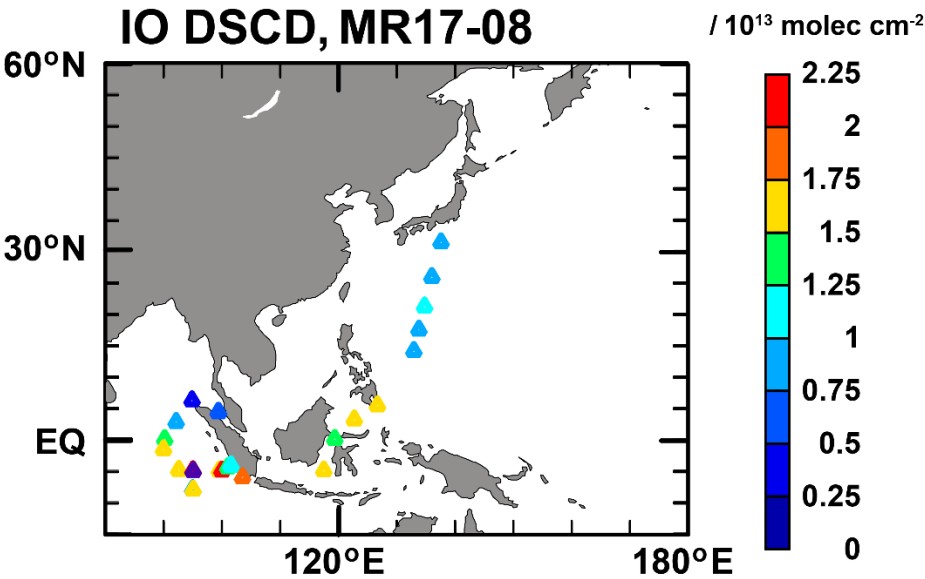

**Figure S6:** Daily median IO content ((DSCD) for an elevation angle of 3°; molecules cm$^{-2}$) observed from the R/V *Mirai* for the MR14-06 (leg1), MR15-04, MR15-05, MR16-06, MR16-09 (leg3), MR17-05C, and MR17-08 cruises.

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
