# Peer review of "Full latitudinal marine atmospheric measurements of iodine monoxide"

_Atmospheric Chemistry and Physics, 2021_

## Author Comment (AC1)

**Reply to Referee #1**

We thank the reviewer for your careful review, constructive comments, and corrections suggested for the manuscript. All changes are highlighted in the track-changes manuscript. A detailed description of our revisions is presented below.

This latter point is well made, however, as it advocates for a rethinking of the representation of iodine fluxes from the ocean it demands a rather high level of scrutiny. The critical illustration is Fig. 3, in which three box-model cases: 1) only $O_3$-dependent iodine fluxes, 2) roughly half dependent half independent fluxes, 3) fully independent fluxes are compared with observations of $O_3$ and IO. While the overall correlation is negative, The Case-1 envelop guides the eye to see that the low $O_3$ observations appear to cluster as two populations which are displaced along the overall negative correlation but individually have positive correlation. All box model cases show consistent behavior for $O_3$ above ~13 ppbv, roughly parallel to the overall correlation, only those with an $O_3$-independent iodine source can reproduce the lowest $O_3$ mixing ratios. However, without offering specific evidence of the $O_3$-independent source other explanations bear consideration. I have the following suggestions for the authors to consider:

1. Is there any specific evidence to support an $O_3$-independent source of iodine?

    1. What measurements of organic iodine fluxes and concentrations are available in the study area, what are the modeled organic fluxes from e.g. Ordóñez et al., (2012)?

    2. For the photooxidation of I⁻ (Watanabe et al., 2019) is there a difference in solar illumination or some other photo-activity proxy between the different observations?

As the referee has pointed out, positive correlation with two populations can be recognized in low $O_3$ and high IO condition in Case 1 in Figure 3. One suitable explanation of the two populations can be done by flux change from wind speed or SST. However, no clear relation was found between IO and wind speed or IO and SST in the two populations (Figure A). Because the HOI flux could be higher in lower wind conditions (MacDonald et al., 2014), the flux change that occurs as a result of wind speed might partly explain the two populations. Actually, high IO concentration was sometimes observed at low wind speed (on Nov. 27, for example; Figure A), but low IO concentration was also observed at low wind speed. Thus, no clear correlation between the two was found in the whole IO-$O_3$ plot.

Another suitable explanation can be Case 2 in Figure 3, in which most of observation points are covered. In this case, the weakened flux is accounted for (as described in the original manuscript, the weakened flux might be explained by dissolved organic carbon (Shaw and Carpenter, 2013) or the presence of a sea-surface microlayer (Tinel et al., 2020) impeding

iodine vaporization). In this case, the added "O$_3$-independent" flux (~$4.8 \times 10^7$ molecules cm$^{-2}$ s$^{-1}$) is not explainable solely by flux from photolysis of iodocarbons (approximately $10^7$ molecules cm$^{-2}$ s$^{-1}$) generally assumed in the 3D models (Saiz-Lopez et al., 2014; Sekiya et al., 2020; Sherwen et al., 2016). While indirectly considering the global total fluxes of CH$_2$IX (X = I, Br, Cl) as described by Ordóñez et al. (2012) in these model simulations, the "Chl-a-based" parameterization reduced the fluxes to too-low levels over this oceanic region. Actually, as previously described in the text, the iodocarbon flux originally reported from a cruise (TransBrom) in the closest oceanic region was even higher, ~$6.81 \times 10^7$ molecules cm$^{-2}$ s$^{-1}$ assuming immediate photolysis from CH$_2$IX (X = I, Br, Cl). Although we have not measured iodocarbons during the studied cruises, we believe that the past results provide good support. We might not need a brand new flux mechanism but rather a good parameterization of the traditional organoiodine fluxes (including their photolysis) over the region. More measurements and parameterizations must be made available for future studies. We added related descriptions to the revised manuscript (P7, L194−196, track-changes manuscript).

*While indirectly considering the global total fluxes of CH$_2$IX (X = I, Br, Cl) as described by Ordóñez et al. (2012) in these model simulations, the "Chl-a-based" parameterization reduced the fluxes to too-low levels over this oceanic region.*

2. Since the points not captured by Case 1 are plausibly only vertically displaced from it, what is the effect of varying the initial O$_3$ mixing ratio? This could be caused by some upwind loss process or else reflect variable entrainment (Kanaya et al., 2019).

Figure B is a scatter plot of O$_3$ and IO for the O$_3$-dependent case, but the initial O$_3$ mixing ratio reduce to 16 ppbv. The covering area is almost equal to that in Figure 3 (18 ppbv); the shift is not purely downward but toward the lower-left corner because the lower initial O$_3$ concentrations assumed here resulted in overall lower flux of inorganic iodine. Therefore, the observed behavior does not appear to be simply explained by the varied initial O$_3$ mixing ratio.

3. The subcases already illustrate the effect of varying the magnitude of the iodine flux, and by extension sea-surface I$^-$, but what about the speciation of the iodine flux (i.e. I$_2$ vs HOI), would changing this change the correlation? There is likely a pH dependence to speciation of the O$_3$-dependent fluxes (e.g. Macdonald et al., (2014); Moreno and Baeza-Romero, (2019)). In addition, since the photooxidation pathway emits I$_2$ (and not HOI) this could offer insight into that hypothesis also.

We made a sensitivity model run with all the flux as I$_2$. However, the changes were only 1% for IO levels and 2% for O$_3$ levels, respectively, with Figure 3. The negative correlation between the two was the same.

4. It seems that the authors have not included heterogeneous reactions which recent studies have suggested have been previously underestimated (Tham et al., 2021), could these impact the trend?

The heterogeneous processes on sea-salt aerosol of Tham et al. (2021) were incorporated into our 3-D model (Sekiya et al., 2020). Sensitivity tests of the heterogeneous uptake coefficient on sea-salt aerosol were done. HOI and other gases were sensitive to the uptake coefficient, but IO was not sensitive to the uptake coefficient. Further research on this point is needed.

[Figure]

**Figure A**: IO mixing ratio observed by MAX–DOAS (pptv) versus *in situ* ozone mixing ratio (ppbv). Colors indicate wind speed (left) and SST (right).

[Figure]

**Figure B**: IO mixing ratio observed by MAX–DOAS (pptv) versus *in situ* ozone mixing ratio (ppbv). Results of box-model simulations with "$O_3$-dependent" emission fluxes of iodine compounds are superimposed as blue same as Figure 3, but initial $O_3$ mixing ratio reduce to 16 ppbv.

Furthermore, it would be helpful if the authors could be more specific in where they expect the posited $O_3$-independent source to be relevant. Is this a feature of the WPWP or relevant across latitudes? Is it possible that there is a less direct influence $O_3$ might play? In particular, studies of ice cores and tree rings (Cuevas et al., 2018; Legrand et al., 2018; Zhao et al., 2019) indicate a roughly threefold increase in iodine since c. 1950 at least ~50% attributed to anthropogenic $O_3$. If half of the inorganic flux were $O_3$-independent as suggested by Case 2, then either some other cause should be searched for, or the change in $O_3$-dependent fluxes to produce the observed change is even more dramatic than previously thought.

Thank you very much for pointing out this important aspect. We added related descriptions to the revised manuscript (P7, L225−229, track-changes manuscript).

*Results of recent studies indicate a roughly threefold increase in iodine since the 1950s, with at least 50% attributed to anthropogenic $O_3$ (Cuevas et al., 2018; Legrand et al., 2018; Zhao et al., 2019). If half of the inorganic flux were $O_3$-independent, as suggested by Case 2, then either some other cause should be sought, or the change in $O_3$-dependent fluxes to produce the observed change is even more dramatic than previously thought.*

Line 160-161: Chlorophyll alone is not enough to exclude an organic iodine source on two counts. Firstly, organic iodine fluxes are not necessarily biotic in origin but might have an abiotic source. Secondly, the mesotrophic conditions characterized by MODIS correspond to those conditions observed to have the largest fluxes of organic iodine in some previous studies e.g. Jones et al., (2010).

We included corresponding descriptions to the revised manuscript (P6, L172, track-changes manuscript).

*In addition, the chlorophyll content, based on satellite MODIS measurements (NASA Level-3 ver. 2018) in the source region, was also low (Figure 5), implying that any organic source of iodine can be expected to be negligible (although we also must consider abiotic organic source as well as mesotrophic conditions (Jones et al., 2010)).*

In addition, the chlorophyll content, based on satellite MODIS measurements (NASA Level-3 ver. 2018) in the source region, was also low (Figure 5), implying that organic sources of iodine would be weak. However, importance of abiotic sources and mesotrophic conditions (Jones et al., 2010) needs to be paid attention. Later we will come back to this point.

 The authors state that there are insufficient data to document diurnal IO variations accurately, however, Fig. 2 indicates good temporal coverage was achieved for some days and it seems evident that there is wealth of IO data more generally. Is there some particular set of data which are missing or something else limiting the retrieval of diurnal variation?

We added a figure of diurnal variation (Figure S4). Although no clear diurnal variation was observed, clear day-to-day variation was observed as shown in Figure 7.

The authors describe an "iodine fountain" in the WPWP which does appear to exist in Fig. 4, however, as the authors acknowledge Fig. 6 shows no clear correlation between SST and IO. The evidence for attributing the fluxes to SST seems at best mixed. For both the WPWP and the Maritime Continent it is clear that there is a lot of variability. Examining the temperature contours it doesn't seem clear that SST would better explain the pattern than latitude. What distinguishes the "fountain" from being a tropical feature of unknown cause from specifically tying it to SST? Relatedly, the authors have described a number of differences between the western Pacific and Atlantic, e.g. higher SST, lower $O_3$. Related to the point above about latitudes, the authors seem to suggest that the "iodine fountain" is a particularity of the WPWP and perhaps maritime continent but not of the Atlantic. But a clearer message on this point would be helpful.

Here we described an "iodine fountain" as a large-scale feature of the WPWP from the global point of view. In detail, as the reviewer pointed out, the one-to-one correlation is not present between SST and IO; however, high IO content was almost always observed at high SST (over approx. 30°C), although the IO content was not always high over high SST area. The causes of the fine-scale features would be studied in the future. To demonstrate this relation, we added Figure S5 in the Supplemental materials. It is noteworthy that Prados-Roman et al. (2015) reported that the highest IO was observed in the western Pacific (in their Figure 4) (in the open ocean).

 "006C" here is presumably "l"

Corrected.

 More recent papers on the $O_3$-dependent iodine source which should be mentioned for offering further consideration of physical and chemical drivers include Inamdar et al., (2020) and Carpenter et al., (2021).

We added a corresponding description to the revised manuscript.

 Inamdar et al., (2020) or else Mahajan et al., (2019) which includes the underlying measurements bear mentioning as more recent measurements of IO on the open ocean.

We added a citation of Inamdar et al. (2020) to the revised manuscript.

Line 88: Is this exposure time the same for all ELs or is this for a specific EL? If the latter the angle should be specified.

The exposure time was fixed for all ELs.

Line 96-101: The version of MMF described in Friedrich et al., (2019) uses Tikhonov regularization rather than optimal estimation for the aerosol retrieval. Was a more recent version used? Could the author provide the version numbers for MMF and VLIDORT?

The version of MMF used in this study is the same as used in Frieß et al. (2019) and Tirpitz et al. (2021) but with adjusted a priori and variance-covariance matrix settings to fit for IO retrieval. It uses VLIDORT v.2.7. We added a corresponding description to the revised manuscript (P4, L99−103, track-changes manuscript).

*The version of MMF used in this study is the same as used in Frieß et al. (2019) and Tirpitz et al. (2021) but with adjusted a priori and variance-covariance matrix settings to fit for IO retrieval. MMF applies the optimal estimation method and uses a two-step approach in which the aerosol profile is first retrieved from $O_4$ DSCDs. Then, the IO profile is retrieved from the IO DSCDs using the earlier retrieved aerosol profile in the forward model. We used VLIDORT (v.2.7) (Spurr, 2006) as the forward model in a pseudo-spherical multiple-scattering setting.*

Line 103-104: These a priori values are presumably the column integrals, this is should be more explicit by e.g. specifying the IO VCD

We specified the description.

Line 104: While Sa is well understood by an expert audience to be the a priori covariance this should be defined for a non-expert audience.

Corresponding text was added to the revised manuscript (P4, L108, track-changes manuscript).

*The a priori covariance matrix $S_a$ for both aerosol and IO retrieval was constructed using the square of 100% of the a priori profile on the diagonal and a correlation length of 200 m.*

Line 123: "they" here is presumably the fluxes, this is not clear.

This point was changed in the revised manuscript.

Line 32: Another recent paper with field evidence for iodine-derived aerosol particles is He et al., (2021)

We added a citation to and a reference of He et al. 2021.

Line 125: Hayase et al., (2010) and Hayase et al., (2012) predate Shaw and Carpenter, (2013) and show similar effects.

We added citations of these reports to the revised manuscript.

Line 141: Some more information on the $O_3$ data filtering would be useful, e.g. is the hourly average a running average or discrete average? What is the typical magnitude or relative magnitude of σ?

The hourly average is a "discrete" average. The typical magnitude of 1σ over the remote ocean was approximately 0.1−0.5 ppbv. We added a corresponding description to the revised manuscript (P5, L148, track-changes manuscript).

*The typical magnitude of 1σ over the remote ocean was approximately 0.1–0.5 ppbv.*

**References**

[revised manuscript text omitted]

---

## Author Comment (AC2)

**Reply to Referee #2**

We thank the reviewer very much for the careful review and constructive comments. All changes are highlighted in the track-changes manuscript. A detailed description of our revisions is presented below.

Section 2.1: An explanation of the a priori choice is needed. Is it appropriate to use an identical a priori given the wide range of regions covered? Similarly, explanations for the choice of aerosol single scattering albedo, asymmetry parameter, and surface albedo should be given, preferably with references.

For this study, the single scattering albedo, asymmetry parameter, and surface albedo were identical to those reported by Großman et al. (2013). We added a corresponding description to the revised manuscript (P4, L113−114, track-changes manuscript).

*Here, the single scattering albedo, asymmetry factor, and surface albedo were used similarly to work presented by Großmann et al. (2013).*

The authors characterize the experimental uncertainty for the dSCDs, but then do not provide any comparable information for the retrieved IO VCDs or mixing ratios in the lowest 200m. The reader needs this information to interpret the level of support provided by the data for the conclusions. Similarly, given the importance placed on the near surface mixing ratios, the authors need to demonstrate the retrieval is sensitive to the near surface mixing ratio particularly given later comments suggesting that a priori selections play a large role in determining the retrieved surface mixing ratios. How large of an effect is this? If it is minimal compared to daily variations, that statement needs more support. This comment suggests the retrieval does not reflect the true atmospheric state, which raises questions about the ability of the authors to quantify the amount of IO present beyond a dSCD. These questions can be answered by showing averaging kernels that reflect the ability to retrieve the IO mixing ratio in the lowest layer (Peak near 1 near the surface with minimal values in other layers). To summarize over the entire data set the authors should provide statistics on the total DOFS for the retrieval as well as the DOFS in the near surface layer. This information will give the reader confidence that the IO values being presented are meaningful, and the papers conclusions are well supported.

The DOFs for the $NO_2$ profile during the MR14-06 (leg1) cruise were 1–1.4. We added a related description of DOFs (P4, L114−115, track-changes manuscript).

*The degrees of freedom (DOFs) for the IO retrieval for MR14-06 (leg1) were 1–1.4.*

Because the vertical profile information was insufficient, the IO concentration near the surface depends on the shape of the *a priori* profile used for the retrieval, as described in the original manuscript (P6, L165−166). However, day-to-day variations near the surface or

correlation between ozone and IO were unaffected by the choice of profile (for example, we retrieved the IO profile using a priori profiles constructed as an exponentially decreasing profile with scale height of 1000 m, as did Großman et al. (2013). In this case, the IO content near surface is lower, but clear negative correlation with ozone was obtained).

Section 2.4

A characterization of the uncertainties associated with the in-situ ozone and CO measurements should be added to this section.

We added a description about uncertainties to the revised manuscript (P5, L150−153, track-changes manuscript).

*The $O_3$ instrument was calibrated twice per year in the laboratory, before and after deployment, using a primary standard $O_3$ generator. The CO instrument was calibrated on board twice per year, on embarking and disembarking of the instrument, using a premixed standard gas. The reproducibility of the calibration was to within 1% for $O_3$ and 3% for CO (Kanaya et al., 2019).*

Line 157. The authors mention insufficient data to show diurnal variations. This statement needs more clarification. You have a lot of data, more than most folks trying to measure IO, why do you not feel good about showing diurnal variations?

We added a figure of diurnal variation (Figure S4). No clear diurnal variation was observed, but clear day-to-day variation was observed as shown in Figure 7.

Line 207: Figure 6 doesn't support the statement of no correlation by itself. I just see timeseries of wind speed and SST, with no attempt to relate these quantities to IO or ozone.

We added the median and $1\sigma$ values to the wind speed and SST similarly to CO in Figure 6 to clarify their mutual correlation. The correlation coefficient between SST and IO was 0.39. That between SST and $O_3$ was −0.51. That between wind speed and IO was −0.45. Also, that between wind speed and $O_3$ was 0.59. It is noteworthy that the correlation coefficient between IO and $O_3$ was −0.75, which is much higher than others, and thus being the dominant feature.

Section 4: Why is it important that IO was detected at low latitudes?

High IO content was observed at low latitude around Indonesia (near the coast; not remote ocean) as well as western tropical Pacific. This sentence might be confusing: we deleted that.

Figure 3: This figure needs error bars on the IO and ozone measurements to show the spread over the data set. I'm also unclear why the linear fit is calculated/shown. I didn't see a reference to it in the text, unless the goal is simply to show anti-correlation, in which case showing an R value makes more sense then the linear fit equation. If there is something important about the fit equation, it would be helpful to know what type of linear fit was done, particularly since the temporal variability of the IO measurements and ozone measurements are not necessarily linked.

As described above, we added an explanation about the uncertainties of $O_3$. We suppose that it would be a natural choice to select a linear fitting line in Figure 4 to show the dominant feature. In accordance with the referee's comment, we added the correlation coefficient to Figure 3 ($R = -0.75$).

Figure 4: Why plot dSCDs rather than VCDs or the surface mixing ratio? dSCDs don't really have much meaning to folks outside the DOAS community. While I find this figure very helpful for showing the cruise tracks and overall spatial extent of the data set, I find myself also wanting to be able to see each cruise plotted individually so I can examine the dataset for each cruise individually. Right now it seems like there are a lot of data points plotted on top of each other. Can you put plots for each cruise in the supplement for the curious reader?

In this study, because we were unable to obtain sufficient vertical information from DSCDs, we showed IO in DSCDs. During the MR14-06 cruise, positive correlation was found between IO VMR and IO DSCDs. Following your comment, we added the DSCDs plots for each cruise in the supplement (Figure S6).

Why are figures 5-7 only shown for 1 cruise?

We specifically intend to examine IO and $O_3$ variations at high SST area in this study and thus chose the MR14-06 Leg 1 cruise here

Data availability: I don't see a data availability statement showing where the data underlying this paper can be obtained, which I believe is a requirement for publication in ACP, and also a generally helpful thing to do for the broader scientific community.

Thank you for your comments. We created a section explaining data availability in the revised manuscript.

Line 34: regiona006C to regional?

This was corrected to regional.

**References**

Großmann, K., Frieß, U., Peters, E., Wittrock, F., Lampel, J., Yilmaz, S., Tschritter, J., Sommariva, R., von Glasow, R., Quack, B., Krüger, K., Pfeilsticker, K., and Platt, U.: Iodine monoxide in the Western Pacific marine boundary layer, Atmos Chem Phys, 13, 3363-3378, 10.5194/acp-13-3363-2013, 2013.

---

## Author Response (AR2)

**Reply to Editor**

Thank you for your considerate handling of our manuscript. According to the referee's comments, we added the requested descriptions to the revised manuscript (P7, L197−198, L222−225). We also added typical IO averaging kernel in Figure S1 as suggested by the referee. We also added some references in Table 2 because one of the co-authors was suggested to referee the original works from which these rates were obtained.

One reviewer recommends minor revisions prior to publication. Please follow the recommendation closely in the revised manuscript.

In addition, the other reviewer made the following suggestions in the event of minor revisions: 1) This sentence from the response "We might not need a brand new flux mechanism but rather a good parameterization of the traditional organoiodine fluxes (including their photolysis) over the region." or something to the same effect would still be useful in the text.

2) The quoted degrees of freedom make clear that the near surface IO might have some interference from the column. If the authors could offer more detail on the vertical distributions of information content it would be better.

**Reply to Referee #2**

We thank the reviewer very much for the constructive comments.

In response to my concerns about the retrievals, the authors have added one line to the manuscript detailing the DOFs of the data set which average slightly above 1. In the response and further in the manuscript, they further explain this piece of information is a surface mixing ratio and information in higher layers is reflective of a priori selection rather than measurement data. This point would be improved with an exemplar averaging kernel showing surface sensitivity (could be in supplement). I'm concerned that the authors still have not presented the uncertainties associated with the retrieved IO mixing ratios. It sounds like the retrieved quantities heavily depend on a priori selection, which raises the question of why the authors are doing a retrieval at all? Presumably the low elevation slant columns would be sufficient to show the relationships described in the manuscript. I understand a mixing ratio is needed for the model comparisons, but if the uncertainty on the retrieved quantity is to high to be useful, one questions the utility of this comparison to begin with.

We added a typical averaging kernel for IO in Figure S1. To test the sensitivity of vertical profile shapes on the retrieved surface IO mixing ratio, a different a priori profile was tested where an exponentially decreasing profile with a scale height of 1000 m (same as Großman et al, 2013), instead of constant below 500 m with an exponentially decreasing profile above 500 m; the IO mixing ratio changed from ~0.8 to ~0.6 pptv. Considering the 3-D model simulation outputs, the tested profile here might be already off the likely range and thus the mixing ratio change should be regarded as maximum. Note that as described in the original manuscript, day-to-day variations near the surface were unaffected by the choice of profile. As such, as the reviewer pointed out, discussion on the basis of DSCD would be preferred, but we used both DSCD and mixing ratio for the comparison with the mixing ratios derived from our 0-D model, with caution.

This information from the response should be incorporated into the manuscript. "The correlation coefficient between SST and IO was 0.39. That between SST and O3 was −0.51. That between wind speed and IO was −0.45. Also, that between wind speed and O3 was 0.59. It is noteworthy that the correlation coefficient between IO and O3 was −0.75, which is much higher than others, and thus being the dominant feature. "

We added a related description to the revised manuscript (P7, L222−225).

Großmann, K., Frieß, U., Peters, E., Wittrock, F., Lampel, J., Yilmaz, S., Tschritter, J., Sommariva, R., von Glasow, R., Quack, B., Krüger, K., Pfeilsticker, K., and Platt, U.: Iodine monoxide in the Western Pacific marine boundary layer, Atmos Chem Phys, 13, 3363-3378, 10.5194/acp-13-3363-2013, 2013.